# CO$_2$ Curing on the Mechanical Properties of Portland Cement Concrete

**Yung-Chih Wang** [1]**, Ming-Gin Lee** [2],*** , Wei-Chien Wang** [1]**, Yu-Cheng Kan** [2]**, Shih-Hsuan Kao** [1]
**and Hsien-Wen Chang** [2]

[1] Department of Civil Engineering, National Central University, Chungli, Taoyuan 320, Taiwan;
wangyc@ncu.edu.tw (Y.-C.W.); a654.joy@gmail.com (W.-C.W.); zxcvbmike@gmail.com (S.-H.K.)

[2] Department of Civil and Construction Engineering, Chaoyang University of Technology,
Taichung 413, Taiwan; yckan@cyut.edu.tw (Y.-C.K.); a22792035@gmail.com (H.-W.C.)

* Correspondence: mglee@cyut.edu.tw

**Abstract:** This study was to evaluate the CO$_2$ curing on mechanical properties of Portland cement concrete. Three different specimen sizes (5 × 10 cm, 10 × 20 cm, and 15 × 30 cm cylinders), three CO$_2$ concentrations (50%, 75%, 100%), three curing pressures (0.2, 0.4, 0.8 MPa), three curing times (1, 3, 6 h), two water cement ratios (0.41, 0.68) for normal and high-strength concretes, and two test ages (3, 28 days) were used for this investigation. Before using the CO$_2$ curing process, the concrete samples reached the initial set at approximately 4 h, and the free water in the samples was gradually removed when dry CO$_2$ gas was injected. The test results show that the 3-day early compressive strength of normal concrete cured by CO$_2$ is higher than that of concrete cured by water, but the difference is not obvious for high-strength concrete cured by CO$_2$. In addition, there is a size effect on the strength of the 5 × 10 cm and 15 × 30 cm cylinders, and the strength conversion factor k$_{s5}$ value obtained for the 28-day compressive strength is greater than 1.18. Compared to conventional water-cured concrete, the elastic modulus of carbon dioxide-cured one generally increases in proportion to the square root of the 28-day compressive strength. It was observed that there are only minor differences in the four $E_C$ empirical equations obtained by CO$_2$ curing from 5 × 10 cm and 10 × 20 cm cylinders, respectively.

**Keywords:** carbon dioxide; compressive strength; modulus of elasticity; rupture modulus; specimen size

## 1. Introduction

CO$_2$ is an important greenhouse gas, and its use in concrete curing may save energy and reduce the carbon content of the atmosphere. The origins of CO$_2$ curing go back decades or centuries, and it is not a new technology. Due to the extremely low concentration of CO$_2$ in the air and its slow diffusion rate, the reaction of CO$_2$ with cement mortar is slow. These result in an insignificant development of the strength of the cement mortar or concrete. In recent years, researchers have tried to explore the carbonization mechanism and its application in the rapid curing of cement materials [1–3]. Other scholars have found that in addition to concrete moisture content, several factors can affect the CO$_2$ curing process, including pressure intensity, pressure time, and curing duration [4,5].

During CO$_2$ curing, calcium silicates such as C$_3$S and C$_2$S in cement are subject to carbonation reactions usually faster than their hydration. Therefore, CO$_2$ curing of fresh concrete will be beneficial to the rapid acquisition of its strength [5]. The carbonization reaction formulas of calcium silicates such as anhydrous alite (C$_3$S, 3CaO SiO$_2$) and belite (C$_2$S, 2CaO SiO$_2$) are shown below [6,7]:

$$3CaO \cdot SiO_2 + (3 - x)CO_2 + nH_2O \rightarrow xCaO \cdot SiO_2 \cdot nH_2O + (3 - x)CaCO_3 \qquad (1)$$

$$\text{and } 2CaO \cdot SiO_2 + (2 - x)CO_2 + nH_2O \rightarrow xCaO \cdot SiO_2 \cdot nH_2O + (2 - x)CaCO_3 \qquad (2)$$

where $xCaO \cdot SiO_2 \cdot nH_2O$ ($C_xSH_n$) refers to the product calcium silicate hydrate, which is simply represented by C-S-H gel. Another carbonation product is calcium carbonate ($CaCO_3$). In addition, the cement hydration product calcium hydroxide will also be carbonated, and the reaction is as follows:

$$Ca(OH)_2 + CO_2 \rightarrow CaCO_3 + H_2O \tag{3}$$

Carbonation of calcium silicate hydration ($C_xSH_n$) gel is expressed as:

$$xCaO \cdot SiO_2 \cdot nH_2O + CO_2 \rightarrow CaCO_3 + (x - 1)CaO \cdot SiO_2 \cdot nH_2O \tag{4}$$

The calcium carbonate produced early is precipitated in the pores of the cement slurry. Therefore, cement-based materials can refine pores, enhancing durability and strength [7,8]. The results from Pingping et al. [9] showed that the calcite formed by initial carbonation was consumed during the hydration reaction of $C_3A$ to form calcium monocarbon aluminate. In addition, $Ca(OH)_2$ was not detected in the reaction of formation of calcium silicate hydrates [9,10]. Concrete carbonation or neutralization is a process in which cement hydration products react with atmospheric carbon dioxide [11,12]. Therefore, concrete structures are no strangers to carbonation occurrence. This is a natural reaction that occurs when concrete is exposed to atmospheric carbon dioxide, called efflorescence carbonation or weathering carbonation. Weathering carbonation is a very slow process because it lowers the pH of the concrete, causing the steel bars in the concrete to corrode. If the above carbonation process is carried out under a controlled environment chamber in the early stage of concrete $CO_2$ curing and strength increase, it is called curing carbonation process [13,14]. Both advantages and disadvantages of concrete $CO_2$ curing are showed in Table 1 [9–29]. It is known from Table 1 that advantages of concrete $CO_2$ curing helps to reduce permeability, porosity and ettringite formation [9,15,16,25–29]. Furthermore, $CO_2$ curing increases resistance to external sodium and magnesium sulfates [17,21,24], acids [16], chloride ion penetration [15,21,22], carbonation weathering [15], drying shrinkage [19,28,29], and freeze–thaw damage [17].

**Table 1.** Advantages and disadvantages of concrete $CO_2$ curing.

| Advantages of Concrete $CO_2$ Curing | Disadvantages of Concrete $CO_2$ Curing |
| --- | --- |
| 1. Fast strength gain.<br>2. A stable solid product is produced due to the carbonization process.<br>3. $CO_2$ is an important greenhouse gas and its use in concrete curing consumes and reduces the carbon content of the atmosphere.<br>4. Reduction of porosity, permeability, and ettringite formation.<br>5. Increased resistance to external sodium and magnesium sulfate, acid attack.<br>6. Increased the resistance to attack by weathering carbonation, damage from freeze–thaw, and drying shrinkage.<br>7. Reduce chloride ions penetration. | 1. The reaction of $CO_2$ with the concrete elements lowers the pH value. Therefore, it may cause corrosion of steel bars in reinforced concrete.<br>2. For precast units only and not suitable for reinforced structure. |

The results from Ravikumar et al. [30] suggest that the net $CO_2$ benefit from carbon capture and concrete utilization is more likely to be negative. That is, in the 99 published experimental datasets, there were 56 to 68 net increases in $CO_2$, depending on the source of $CO_2$. This is a promising strategy to increase the net $CO_2$ benefit from carbon capture and utilization of concrete by curing through $CO_2$ and increasing the compressive strength, and can reduce the curing time and electricity used in curing. In addition, studies have investigated the size effect on the compressive strength of $CO_2$-cured concrete. If a $10 \times 20$ cm cylindrical mold is used, the strength obtained for $CO_2$-cured concrete in

the 20 to 60 MPa range is expected to be 5% higher than that obtained with a 15 × 30 cm cylindrical mold [31]. Other studies have found that in the mid-strength range, such as 20 to 60 MPa, it is practically acceptable to assume equal strength for 10 × 20 cm and 15 × 30 cm molds; the rationale for this assumption must be determined or revised by the standards authority [32,33]. However, some argue that the standard deviations are sufficiently different and that a 15 × 30 cm cylinder twice as large as 10 × 20 cm is required to maintain the same degree of accuracy [33,34], exploring the relationship between different cylinder sizes and mechanical properties using data obtained in the literature [34–37]. The two different cylinder sizes for ordinary strength concrete (≤40 MPa) did not result in test variability or differences in test data, including tests for compressive strength and static and dynamic elastic moduli. However, it has been observed that the size effect becomes significant in concrete above 40 MPa [35]. The aim of this study is to evaluate the effect of $CO_2$ curing on the mechanical properties of Portland cement concrete, and also to explore the different strengths and different specimen sizes of concrete. The scope of the research will be to carry out $CO_2$-curing experiments for normal strength concrete and high strength concrete. The relevant parameters are as follows: (1) water–cement ratio, (2) $CO_2$-curing opportunity, (3) $CO_2$-curing concentration, (4) $CO_2$-curing pressure, (5) $CO_2$-curing time and schedule, and (6) size effect. Finally we found out the optimal values of concrete $CO_2$-curing and recommendations for parameters: concentration, pressure, and duration.

## 2. Experiments

This experiment includes the preparation of materials and equipment, the mix proportion of concrete, the combination of factors, the $CO_2$-curing test, compressive strength, elastic modulus, modulus of rupture, thermogravimetric analysis, and XRD analysis tests.

### 2.1. Concrete Materials and Mix Proportion

The concrete materials used in the experiments consisted of Type I Portland cement produced by Taiwan Cement Corporation and compliant with ASTM C 150 [38], coarse aggregates (crushed stone) ranging from 9.5 mm to #4 sieves, fine aggregates, and water. Aggregates in Taiwan contain large amounts of sandstone, slate, and shale. The maximum particle size of the coarse aggregate is 9.5 mm and its gradation meets ASTM C33 requirements. Among them, the fine aggregate is the river sand in eastern Taiwan. Table 2 shows the mix design for two concrete mixtures. The mix design criteria used in this study were specified in accordance with ACI 211.1 [39] for selecting the proportions of ordinary concrete, heavy concrete, and mass concrete. Normal concrete (No. 1) has a water–cement ratio of 0.68 and a slump of 15 ± 2.5 cm. High-strength concrete (No. 2) has a water–cement ratio of 0.41 and a slump of 10 ± 2.5 cm. The mixing batch of 0.2 cubic meters is determined according to the experimental quantity and the mixing volume of the 0.3 cubic meter concrete mixer. Each batch will produce 20 cylinders of 5 × 10 cm, 20 cylinders of 10 × 20 cm, and 12 cylinders of 15 × 30 cm, with an estimated additional 15% safety reserve.

**Table 2.** Mix proportion for two concretes (kg/m³).

| Mix Proportion | W/C | Cement | Aggregate | Sand | Water |
|---|---|---|---|---|---|
| Normal concrete | 0.68 | 302 | 885 | 1021 | 205 |
| High-strength concrete | 0.41 | 500 | 885 | 881 | 205 |

### 2.2. Concrete Samples

Three different sizes (5 cm diameter × 10 cm height, 10 cm diameter × 20 cm height, and 15 cm diameter × 30 cm height) were prepared according to ASTM C39/39M [40] and used to evaluate the effect of $CO_2$ curing on mechanical properties of Portland cement concrete. Two sizes of cylinders (5 × 10 cm, 10 × 20 cm) were used to study the modulus of elasticity of $CO_2$-cured concrete. The purpose of this study was to evaluate the effect of

cylinder size on the compressive strength of $5 \times 10$ cm, $10 \times 20$ cm, and $15 \times 30$ cm cylinders made from the same batch of concrete cured by carbon dioxide. The main variables identified as likely to affect the strength are cement type, water cement ratio, aggregate type, cement content, admixture, age of testing, and curing method [41]. Therefore, in the experimental phase of this study, different water cement ratio, curing methods, and test ages were selected. The mechanical properties of Portland cement concrete, such as compressive strength, modulus of rupture, and elastic modulus, were selected for quality control testing and quality evaluation.

### 2.3. Concrete $CO_2$ Curing Test

The concrete $CO_2$ curing test device includes a $CO_2$ gas tank, a mixing air tank, a pressure curing chamber, a vacuum pump, a pressure gauge, a regulator, a safety valve, a heater, and a thermometer. The $CO_2$ curing systems have both a mixed air tank and a $CO_2$ gas tank, so the gas concentration can be diluted to simulate the $CO_2$ emitted by the factory, for example, to assist in the transformation of low-carbon production in thermal power plants or cement plants and reduce the carbon emissions of production processes. The $CO_2$ gas was initially supplied to the curing chamber by a gas regulator controlled to supply pressures of 0.2, 0.4, or 0.8 MPa, respectively [34].

Two concrete curing methods were used in this study, $CO_2$ curing and water curing. The $CO_2$ curing process is to place the newly poured concrete specimen for about four hours to reach the initial setting, and then put the specimen into a special pressure chamber for $CO_2$ curing after the mold is removed. Use a gas-tight steel container of approximately 216 L of $60 \times 60 \times 60$ cm cubes as the $CO_2$ curing chamber, as shown in Figure 1, which is evacuated to $-0.01$ MPa before injecting the $CO_2$ gas. The $CO_2$ pressure in the chamber was controlled by a gas regulator with pressure variables of 0.2, 0.4, and 0.8 MPa and curing time variables of 1, 3, and 6 h. The concrete $CO_2$ curing chamber is placed in a room with a temperature of $25 \pm 1$ °C. The $CO_2$ curing chamber test capacity is twenty $5 \times 10$ cm cylinders, twenty $10 \times 20$ cm cylinders, and twelve $15 \times 30$ cm cylinders at a time. The combination of factors used for the concrete $CO_2$ curing test is shown in Table 3 [35].

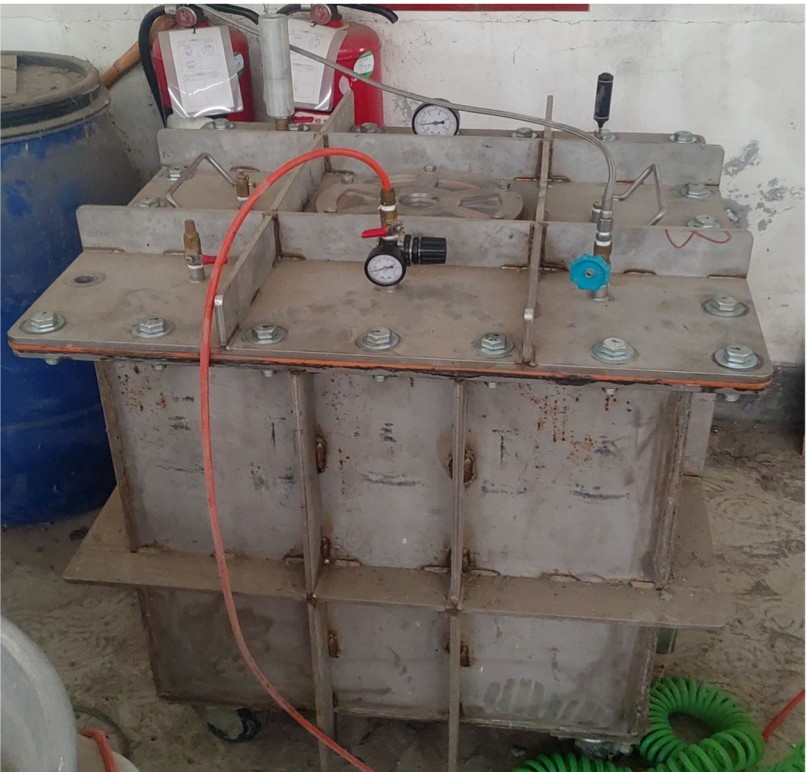

**Figure 1.** $CO_2$ pressure curing chamber.

**Table 3.** Combination factors of concrete $CO_2$ curing test.

| Water–Cement Ratio | Pressure (MPa) | CO$_2$ (%) | Time (Hour) |
|---|---|---|---|
| Normal concrete (0.68)<br>High-strength concrete (0.41) | 0.2<br>0.4<br>0.8 | 50<br>75<br>100 | 1<br>3<br>6 |

Notation

An identification system for specific specimen was used to keep track of the data in this experiment. Cylinders are identified in the order of compressive strength (L or H), $CO_2$ concentration (%), cure time (hour), and cure pressure (0.1 MPa). For example, the cylinder identified as L-50-6-2 is a cylinder made of normal strength concrete with 50% $CO_2$ concentration, cured for 6 h at 0.2 MPa pressure. CL and CH represent normal strength and high strength specimens cured in water, respectively [35].

## 3. Experimental Results

### 3.1. Effect of CO$_2$ Curing on Concrete Compressive Strength

The standard sample size for acceptance testing of compressive strength of concrete in general public works is a 15 × 30 cm cylinder [41]. AASHTO, ASTM, BS, or CSA test standards allow 5 × 10 cm cylinders, 10 × 20 cm cylinders, or 12 × 24 cm cylinders. AASHTO, ASTM, BS, or CSA standards for coring testing allow the use of smaller specimens such as a 5 × 10 cm cylinder, a 7.5 × 15 cm cylinder, or a 10 × 20 cm cylinder. However, these smaller cylinders are not officially used because of the variability and uncertainty in their strength compared to standard size samples (15 × 30 cm cylinders) made from the same tray of fresh concrete. This section of the study attempts to correlate the strength between a standard size specimen (15 × 30 cm cylinder) and a smaller cylinder size. The 3-day and 28-day normal concrete compressive strength results obtained from 5 × 10 cm, 10 × 20 cm, and 15 × 30 cm cylinders by 100% $CO_2$ curing are showed in Figures 2 and 3. Each strength value is the average of the compressive strengths of three test specimens. Figure 2 shows that the 3-day compressive strength of the three cylindrical sizes of normal concrete cured with 100% $CO_2$ is higher than that of conventional water-cured concrete, and the L-100-6-4 specimen exhibits higher strength values. It can be seen from Figure 3 that the 28-day compressive strength of 100% $CO_2$-cured concrete is close to that of conventional water-cured concrete. The experimental data of normal strength concrete through water curing show that there is no difference in the 3-day strength of concrete cylinders of 5 × 10 cm, 10 × 20 cm, and 15 × 30 cm, but the strength value of the 15 × 30cm cylinder at 28 days is lower.

Table 4 shows the ranking of the top three compressive strengths of the three sizes of normal concrete cured with $CO_2$ at each age and the comparison of the compressive strength of the water-cured specimens in the control group (CL). It is found that the higher compressive strength of the $CO_2$-cured specimen is not the combination of 100% $CO_2$ concentration, high pressure, and long curing time, but the combination of 50% concentration and medium pressure. The curing combination of 50-3-2 and 50-1-2 both won 3 first places in compressive strength, indicating that the $CO_2$ pressurized 0.1 to 0.3MPa concentration of 50%, and only 2 h of curing can get better compressive strength.

**Table 4.** Top 3 compressive strengths of all ages of normal concrete cured with $CO_2$ (unit: MPa).

| Age of Hardening | Specimen ID | Φ5 × 10 | Specimen ID | Φ10 × 20 | Specimen ID | Φ15 × 30 |
|---|---|---|---|---|---|---|
| 3-day Strength | 50-1-2<br>100-6-4<br>100-1-2<br>CL | 16.82<br>15.41<br>15.35<br>12.34 | 100-6-4<br>100-1-2<br>75-3-4<br>CL | 15.40<br>14.66<br>14.19<br>12.49 | 50-1-2<br>50-1-4<br>100-6-4<br>CL | 16.15<br>15.27<br>15.00<br>12.59 |
| 7-day Strength | 100-6-2<br>50-3-2<br>100-6-4<br>CL | 22.93<br>22.20<br>21.97<br>19.33 | 50-3-2<br>100-1-2<br>75-6-2<br>CL | 24.24<br>21.81<br>21.12<br>15.90 | 50-3-2<br>50-1-2<br>100-1-2<br>CL | 22.61<br>22.59<br>21.55<br>16.42 |

**Table 4.** *Cont.*

| Age of Hardening | Specimen ID | Φ5 × 10 | Specimen ID | Φ10 × 20 | Specimen ID | Φ15 × 30 |
|---|---|---|---|---|---|---|
| 28-day Strength | 50-1-2 | 36.05 | 75-6-2 | 29.45 | 50-3-2 | 30.15 |
|  | 50-3-2 | 34.23 | 50-6-2 | 28.95 | 50-3-4 | 29.83 |
|  | 75-1-4 | 32.22 | 50-3-2 | 28.94 | 50-1-2 | 28.71 |
|  | CL | 31.58 | 75-3-2 | 27.65 | 100-1-2 | 26.73 |
| 90-day Strength | L-50-1-2 | 40.95 | L-50-3-2 | 36.62 | L-50-1-2 | 33.52 |
|  | L-100-1-2 | 39.16 | L-100-6-2 | 36.49 | L-100-6-2 | 33.26 |
|  | L-100-6-2 | 38.49 | L-75-3-4 | 36.13 | L-100-1-2 | 33.22 |
|  | CL | 34.06 | CL | 30.02 | CL | 30.12 |

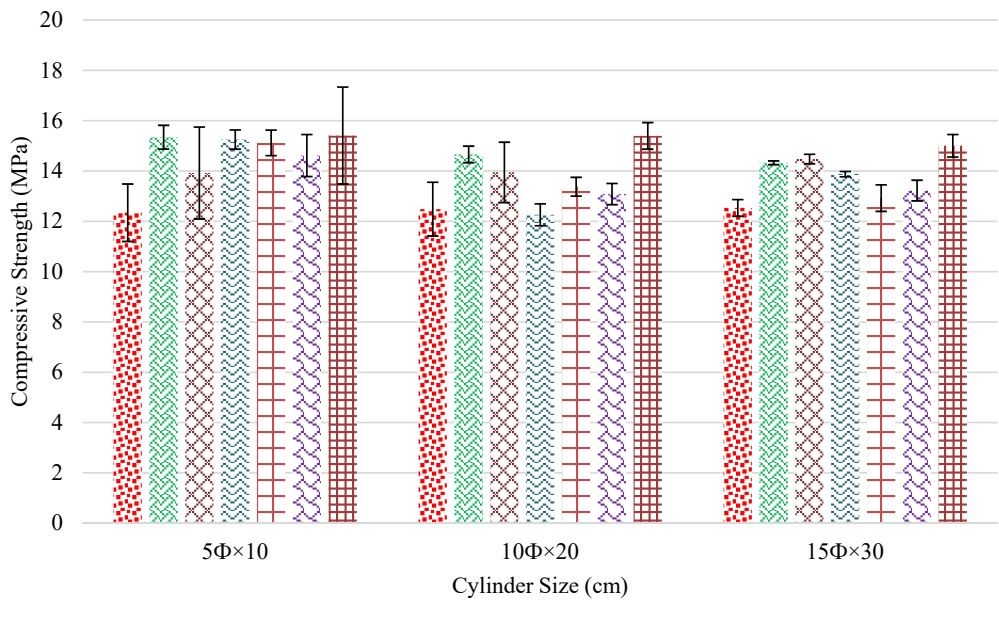

**Figure 2.** Result of 3-day compressive strength of normal concrete of different size cylinders.

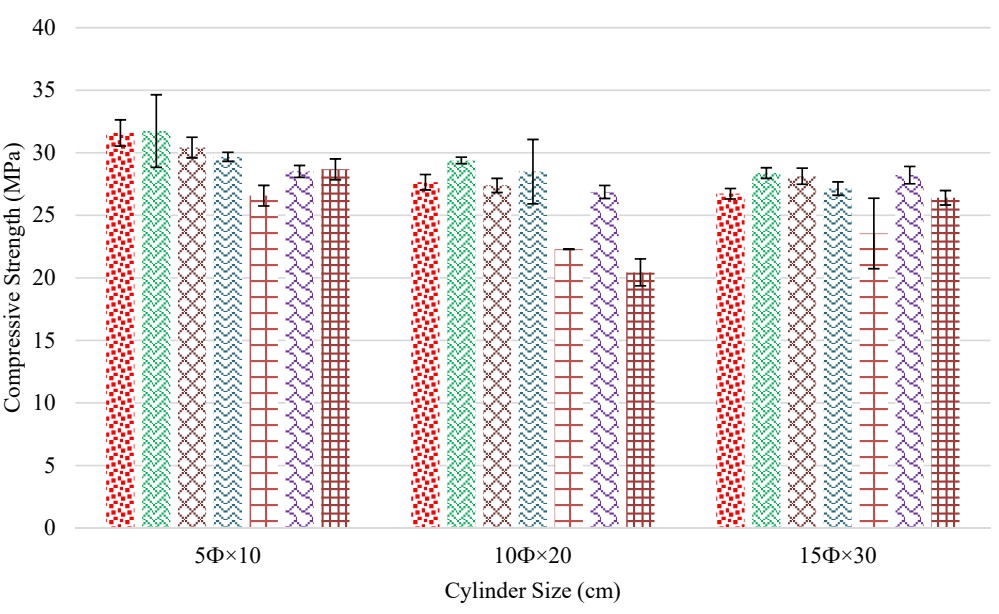

**Figure 3.** Result of 28-day compressive strength of normal concrete of different size cylinders.

Figures 4 and 5 show the 3-day and 28-day high-strength concrete results obtained for $5 \times 10$ cm, $10 \times 20$ cm, and $15 \times 30$ cm cylinders cured by 100% $CO_2$ for 6 h. There is a $5 \times 30$ cm cylindrical specimen in Figure 4, identified as H-100-6-2, which was made of high-strength concrete cured for 6 h at 100% $CO_2$ concentration under 0.2 MPa pressure, and its early 3-day compression strength value is the highest. Figure 5 shows that the the average compressive strength of the three cylindrical size high-strength concretes at 28 days cured with 100% $CO_2$ are close to those of conventional water-cured concrete. The experimental data of high-strength concrete through water curing show that there is no difference in the 3-day strength of concrete cylinders of $5 \times 10$ cm and $15 \times 30$ cm, but the strength value of the $15 \times 30$ cm cylinder at 28 days is lower.

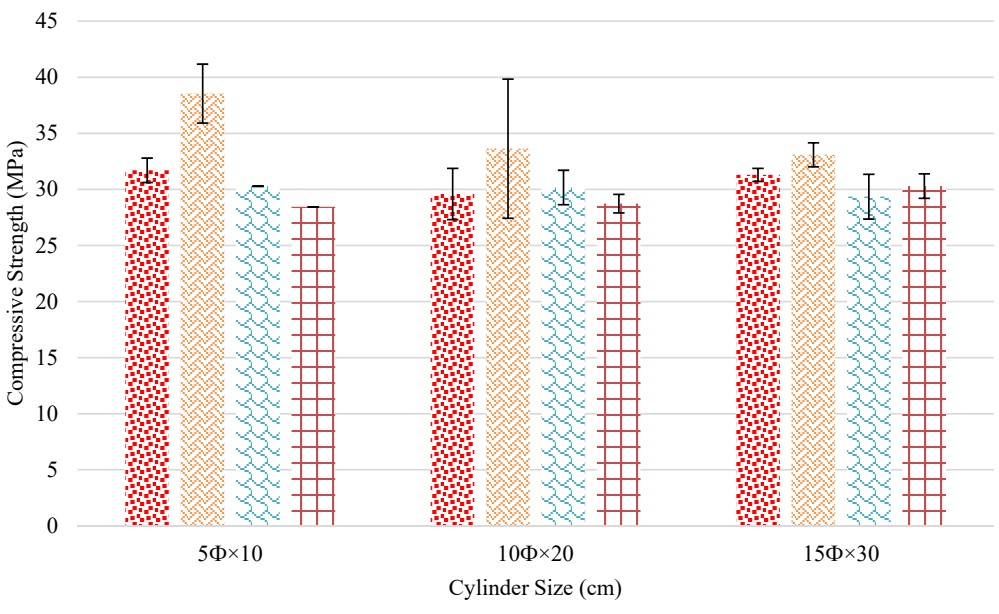

**Figure 4.** Result of 3-day high strength concrete of different size cylinders cured with 100% $CO_2$.

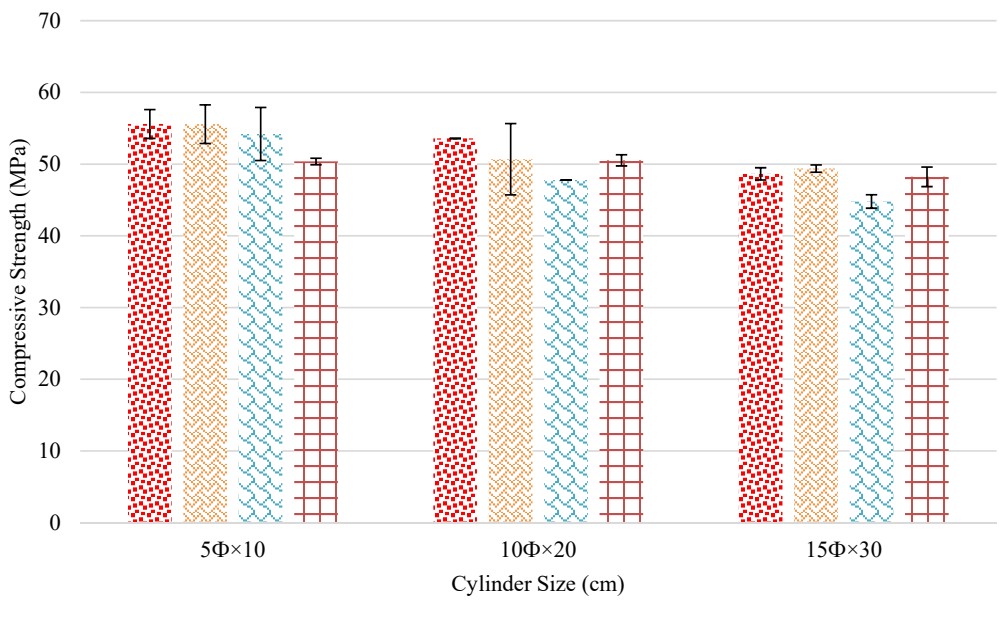

**Figure 5.** Result of 28-day high strength concrete of different size cylinders cured with 100% $CO_2$.

This section investigates the conversion of strength factors between standard cylindrical specimen dimensions of concrete (15 × 30 cm) and smaller dimensions. The compressive strength data and strength conversion factors (ks values) received by water curing for 3 days, 7 days, 28 days, and 90 days from three cylindrical specimen sizes (5 × 10 cm, 10 × 20 cm, and 15 × 30 cm) are shown in Table 5. Each compressive strength value is an average of three cylinder strengths. There was no difference in 3-day, 7-day, 28-day, or 90-day compressive strength between 10 × 20 cm and 15 × 30 cm cylinders for normal strength concrete, with ks ranging between 0.97 and 1.03. It is clear that the 5 × 10 cm cylinder of normal strength or high strength concrete has higher strength at each age than the 10 × 20 cm and 15 × 30 cm cylinders. Additionally, there is a large difference in strength between 5 × 10 cm and 15 × 30 cm cylinders, with ks values up to 1.18 [34].

**Table 5.** Strength and $k_s$ value from different size cylinders (unit:MPa).

| Type of Concrete/Age | Φ5 × 10 cm | Φ10 × 20 cm | Φ15 × 30 cm | $k_{s5}(f_{c5}/f_{c15})$, $k_{s10}(f_{c10}/f_{c15})$ |
|---|---|---|---|---|
| Normal concrete/3 day | 12.34 | 12.49 | 12.53 | 0.98, 0.99 |
| Normal concrete/7 day | 19.33 | 15.90 | 16.42 | 1.18, 0.97 |
| Normal concrete/28 day | 31.59 | 27.65 | 26.73 | 1.18, 1.03 |
| Normal concrete/90 day | 34.06 | 30.02 | 30.12 | 1.13, 1.00 |
| High strength concrete/3 day | 31.70 | 29.59 | 31.29 | 1.01, 0.95 |
| High strength concrete/7 day | 45.80 | 41.46 | 41.72 | 1.10, 0.99 |
| High strength concrete/28 day | 55.60 | 53.58 | 48.64 | 1.14, 1.10 |
| High strength concrete/90 day | 65.43 | 53.78 | 56.20 | 1.16, 0.96 |

*3.2. Effect of $CO_2$ Curing on Concrete Modulus of Elasticity*

The compressive strength and elastic modulus of concrete *Ec* are of great significance for the design of concrete structures and the assessment of the current condition of old concrete structures. Figure 6 was an elastic modulus of concrete cylinder setup with two strain rings. Figures 7 and 8 show the relationship between the modulus of elasticity and the square root of the compressive strength for ordinary or high-strength concrete obtained from 5 × 10 cm and 10 × 20 cm cylinders cured by water or $CO_2$, respectively. Figure 7 was a *Ec* results obtained from 5 × 10 cm cylinders and Figure 8 was *Ec* results obtained from 10 × 20 cm cylinders. The variable in two Figures was largely the $CO_2$ curing type, as the 5 × 10 cm and 10 × 20 cm cylinders were very similar. By collecting and analyzing these test data, a formula for estimating the elastic modulus of concrete in Taiwan was proposed. We multiplied the elastic modulus prediction formula for ACI 318 by a correction factor of 0.8 [42]. That is, Figures 7 and 8 show that the best trend line $Ec = 3750 \sqrt{Fc}$ with the R-squared being about 0.60. It indicates a moderate or good correlation between elastic modulus and square root of strength. Compared with conventional water-cured concrete, the elastic modulus of carbon dioxide-cured concrete also increases continuously in proportion to the square root of the compressive strength. Choosing the most appropriate type of $CO_2$ curing for fresh concrete will have a significant impact on the modulus of elasticity. This highlights the greater sensitivity of $CO_2$ cured concrete that in many cases is not considered within empirical relationships that predict elastic modulus.

In this study, the standard deviation (σ) and the coefficient of variation (COV, the standard deviation divided by the mean value) were used as a means to evaluate the experimental variability of the elastic modulus of concrete. ACI 318 committee [43] suggests an empirical equation that relates *Ec* and *Fc* (*Fc* less than 38 MPa):

$$E_{\text{ACI 318}} = 0.043wc \cdot 1.5\sqrt{Fc} \text{ (MPa).} \tag{5}$$

ACI 363 committee [44] suggests a different equation for linking *Ec* and *Fc* (*Fc* between 21 MPa and 83 MPa):

$$E_{\text{ACI 363}} = (wc\ 2300) \cdot 1.5 \cdot (3320\sqrt{Fc} + 6900) \text{ (MPa).} \tag{6}$$

Euro International CEB-FIP committee proposes an empirical equation relating *Ec* and *Fc* for all concretes:

$$E_{\text{CEB-FIP}} = 22{,}000 \sqrt[3]{(F_c/10)} \cdot (\text{MPa}). \tag{7}$$

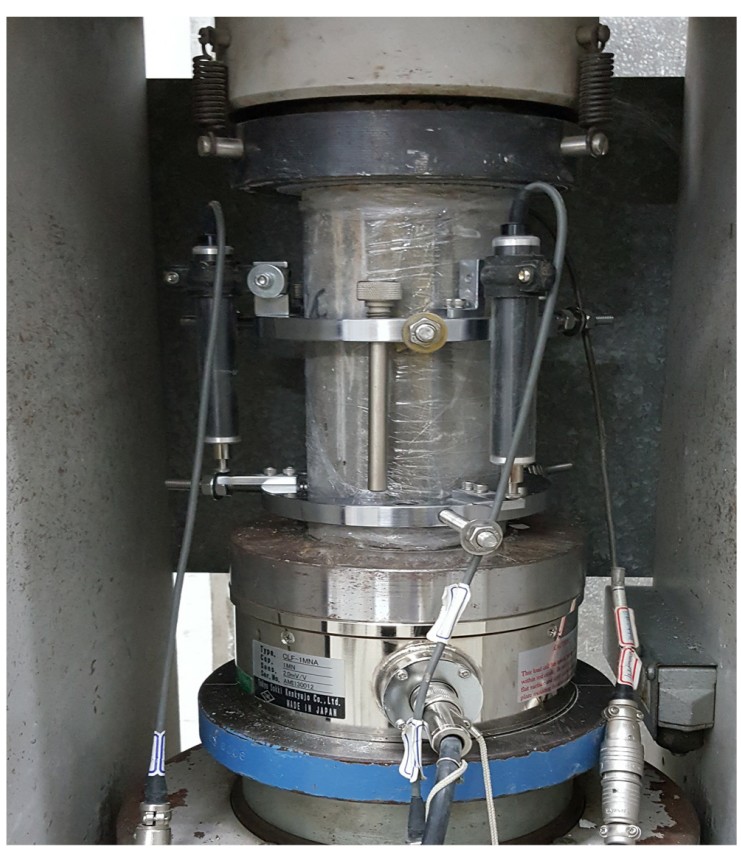

**Figure 6.** Elastic modulus of concrete cylinder setup with two strain rings.

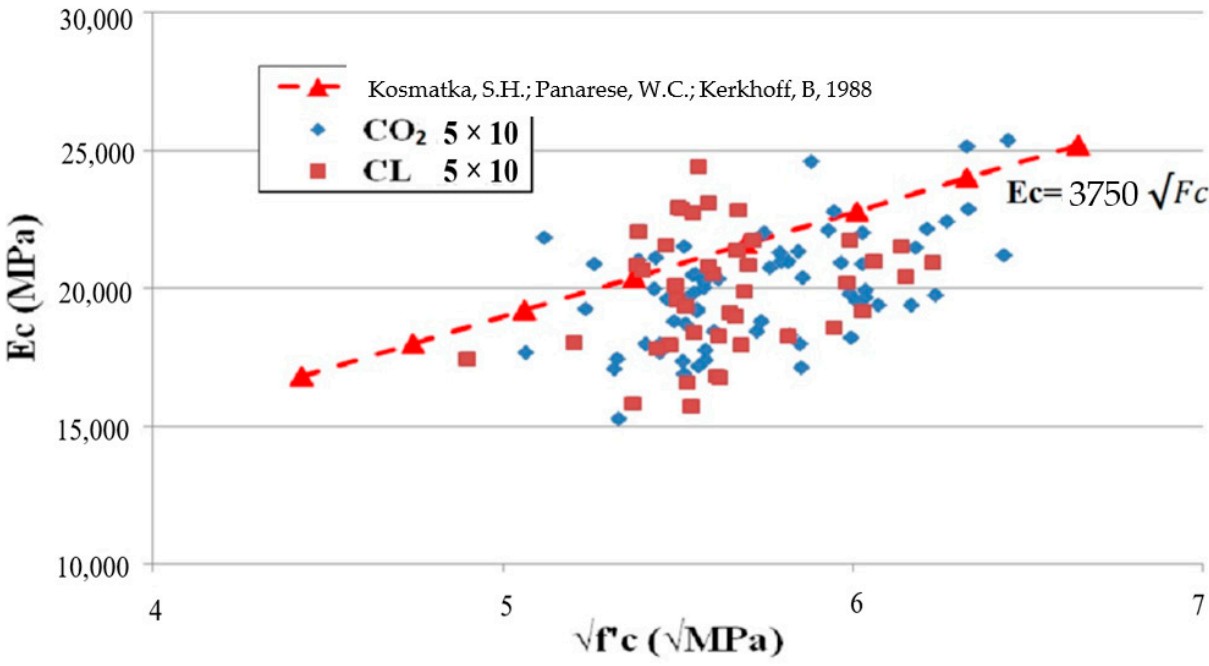

**Figure 7.** Relationship between elastic modulus and square root of strength obtained from 5 × 10 cm cylinders. Partialy adapted from Ref. [42].

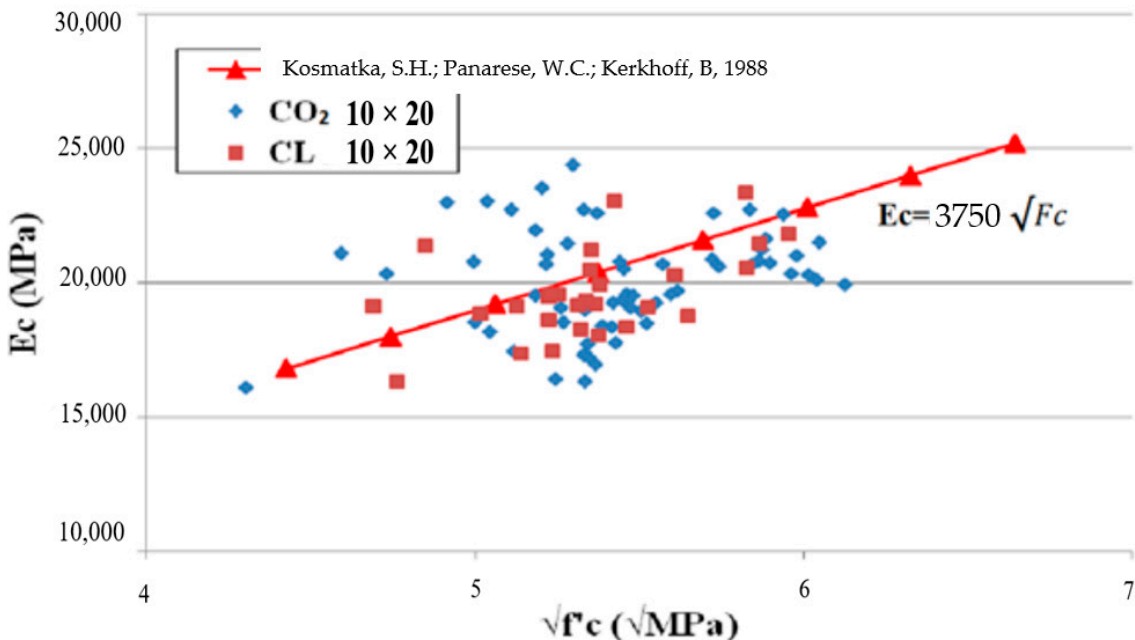

**Figure 8.** Relationship between elastic modulus and square root of strength obtained from 10 × 20 cm cylinders. Partialy adapted from Ref. [42].

When the equations adopted by ACI committee were developed, there was no standard test method to determine *Ec*, so there was a big difference in taking the initial, tangential, or secant modulus according to the definition of the elastic modulus of concrete. Furthermore, the elastic modulus Equations (5)–(7) do not consider key parameters other than compressive strength, such as aggregate unit weight, fibers, mineral admixtures, chemical admixtures, and specimen dimensions [37]. Table 6 compares the *Ec* empirical equation and statistical parameters (COV and σ) of test results obtained from 5 × 10 cm and 10 × 20 cm cylinders by $CO_2$ or water curing. It is observed that there is only a minor difference in four *Ec* empirical equations obtained from 5 × 10 cm or 10 × 20 cm cylinders by $CO_2$ or water curing, whereas the 10 × 20 cm cylinders obtain larger elastic modulus coefficients. The mean COVs for the elastic modulus of concrete for 5 × 10 cm and 10 × 20 cm cylinders were 10.1% and 9.2%, respectively.

**Table 6.** Comparison of *Ec* empirical equation and statistical parameters of $CO_2$ cured concrete.

| Cylinder Size + Curing | Empirical Equation | Standard Deviation | COV (%) |
|:---:|:---:|:---:|:---:|
| $\Phi_{5 \times 10\,cm}$ + $CO_2$ Cured | Ec = 3498.5$(f'_c)^{0.5}$ | 322.8 | 9.2 |
| $\Phi_{5 \times 10\,cm}$ + $H_2O$ Cured | Ec = 3536.6$(f'_c)^{0.5}$ | 385.3 | 10.9 |
| $\Phi_{10 \times 20\,cm}$ + $CO_2$ Cured | Ec = 3704.8$(f'_c)^{0.5}$ | 405.8 | 10.9 |
| $\Phi_{10 \times 20\,cm}$ + $H_2O$ Cured | Ec = 3673.1$(f'_c)^{0.5}$ | 273.6 | 7.5 |
| ACI 318 empirical equation | $E_c = 3750\,(f'_c)^{0.5}$ recommend for use in Taiwan. | | |

### 3.3. Effect of $CO_2$ Curing on Rupture Modulus

In the past, the literatures related to carbon dioxide curing cement were all focused on the compressive strength, and rarely on the tension of the concrete. Therefore, this test is a preliminary study on the rupture modulus of the concrete. The test method for determining the flexural strength of concrete used a simple beam with third point loading according to standard ASTM-C78 [45]. We tested, measured, and recorded the load of the standard beam at failure and calculated the rupture modulus R according to the formula provided by the specification. The higher the rupture modulus, the higher the tensile strength of the

concrete of the specimen. The ASTM-C78 specification proposes a formula relating rupture modulus R, rupture modulus coefficient K and *Fc* as follows:

$$R = \frac{PL}{bd^2} = k\sqrt{f'_c} \qquad (8)$$

In this experiment, there are four 15 × 15 × 53 cm standard beams and three 15 × 30 cm cylindrical specimens in each group. The carbon dioxide curing conditions were 100% carbon dioxide concentration, curing for 6 h and pressurized pressure of 0.4 MPa. After the $CO_2$ curing was completed, it was placed in water for subsequent hydration until the rupture modulus test began at 28 days of age. Figure 9 was a rupture modulus of high strength concrete beam setup.

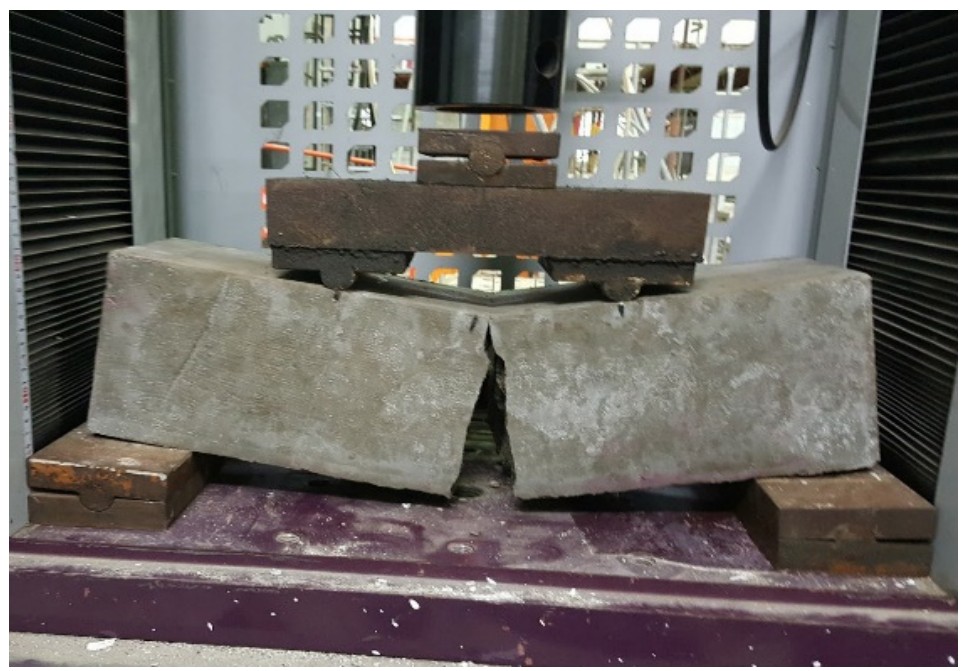

**Figure 9.** Rupture modulus of high strength concrete beam setup.

Table 7 shows the results of this test calculated according to the ASTM-C78 standard method. It is found that the difference between the rupture modulus coefficients of carbon curing and water curing is very small, indicating that the concrete after carbon curing will not increase or reduce the tensile strength due to the formation of calcium carbonate on the surface of concrete. The k value of the rupture modulus of the ACI-318 specification is 2, and the k value obtained through the test is 2.9. Therefore, it can be found that the specification underestimates the concrete tensile force, and the k value is relatively conservative [46].

**Table 7.** Rupture modulus of high strength concrete with $CO_2$ curing and water curing.

| Specimen ID | P | L | b | d | R (Rupture Moduli) | f′c | K | √f′c | K(Average) |
|---|---|---|---|---|---|---|---|---|---|
| CH (water curing) | 4505.2 | 50 | 15 | 15 | 6.54 | 45.55 | 3.10 | 6.75 | 2.92 |
| | 3984.38 | | | | 5.79 | | 2.74 | | |
| | 4183.26 | | | | 6.08 | | 2.88 | | |
| | 4293.79 | | | | 6.23 | | 2.95 | | |
| H-100-6-4 ($CO_2$ curing) | 3972.82 | 50 | 15 | 15 | 5.77 | 46.17 | 2.71 | 6.79 | 2.95 |
| | 3714.49 | | | | 5.40 | | 2.54 | | |
| | 4733.96 | | | | 6.88 | | 3.23 | | |
| | 4888.19 | | | | 7.10 | | 3.34 | | |

Unit: P (kgf), L (cm), b (cm), d (cm), R (MPa), f′c (MPa), K value (rupture modulus coefficient).

### 3.4. Thermo-Gravimetric Analysis

The surface of the $5 \times 10$ cm cylindrical specimen was brushed and cleaned and placed in an oven to prevent moisture. After drying, we cut the middle part of the $5 \times 10$ cm cylindrical specimen to a thickness of about 1 cm. All broken pieces were ground and passed through a No. 100 sieve, and the powder samples were then subjected to thermogravimetric analysis (TGA). The TGA method can examine different hydrates and carbonates. The results of TGA are shown in Figure 10 and Table 8. The mass loss occurs in three main stages during the TGA process: dehydration due to the dissipation of bound water (105–450 °C), dehydroxylation due to decomposition of $Ca(OH)_2$ (450–550 °C), and dehydration due to calcite decomposition and decarburization (550–900 °C). At 90 days of age after carbon curing, the most significant variation between batches was the substantial mass loss that occurred in carbonated samples above 600 °C, which corresponds exactly to the decomposition of the $CaCO_3$ present in the system. This indicates that a large amount of $CO_2$ is captured in the carbonic acid mixture. Comparing carbon curing specimens, it can be found that the higher the curing pressure, the higher the weight loss. When L-50-6-4 (or L-50-6-2) is compared with L-100-6-4 (or L-100-6-2), it can be found that those specimens with lower carbon dioxide concentration have higher weight loss. The $CO_2$ uptake in the carbonate samples is vaguely visible from the TGA results, with a significant increase in carbonate content. At 90 days, both bound water and $Ca(OH)_2$ contents of the carbonate samples decreased significantly, indicating the conversion of C-S-H and $Ca(OH)_2$ to carbonate [47].

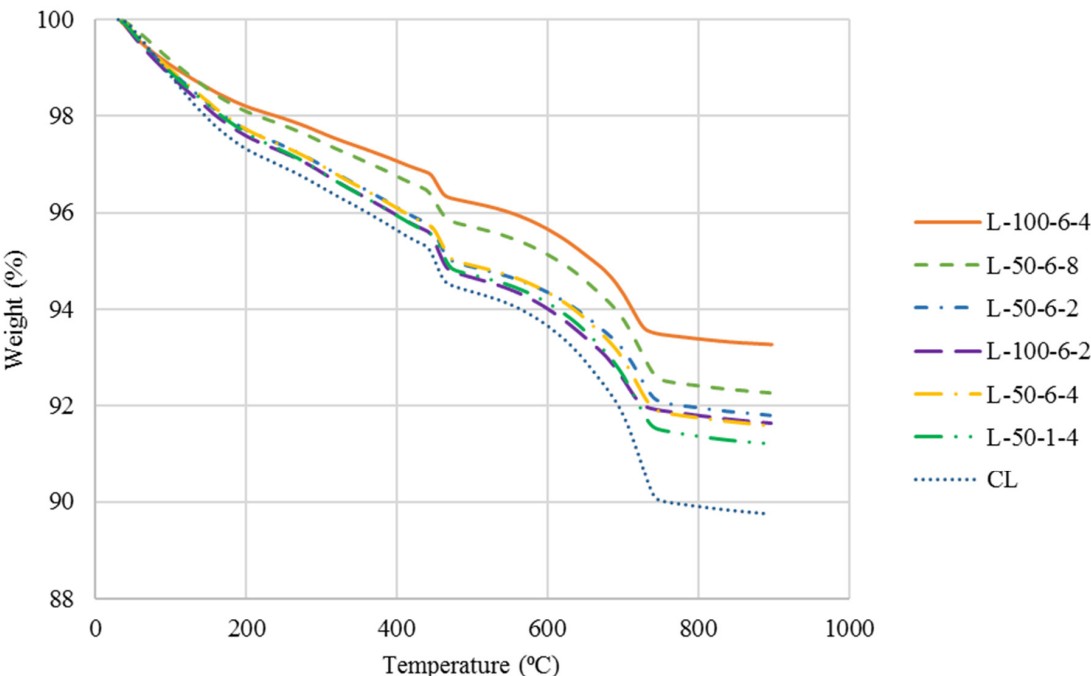

**Figure 10.** TGA curves for water and $CO_2$ cured concrete samples.

**Table 8.** Mass loss (%) for water and $CO_2$ cured concrete over different temperature ranges.

| Specimen ID | 650 °C (%) | 900 °C (%) | 650~900 °C (%) |
|---|---|---|---|
| CL (water curing) | 92.92 | 89.74 | 3.18 |
| L-50-1-4 | 93.54 | 91.21 | 2.33 |
| L-50-6-8 | 94.59 | 92.27 | 2.32 |
| L-50-6-4 | 93.8 | 91.58 | 2.22 |
| L-50-6-2 | 93.86 | 91.8 | 2.06 |
| L-100-6-4 | 95.13 | 93.27 | 1.86 |
| L-100-6-2 | 93.41 | 91.64 | 1.77 |

### 3.5. XRD Analysis of Concrete Specimen

Dicalcium silicate and tricalcium silicate in cement react with carbon dioxide to form C-S-H colloid and calcium carbonate, and the reactions are shown in Formulas (1) and (2). In order to understand whether the concrete will produce calcium carbonate with different crystalline phases after carbon dioxide curing, such as aragonite ($CaCO_3$), vaterite ($CaCO_3$), and calcite ($CaCO_3$), this test uses XRD for qualitative analysis. The above three items and calcium hydroxide (Portlandite, $Ca(OH)_2$) were compared.

After the specimen curing age reaches 90 days, the surface of the $5 \times 10$ cm cylindrical specimen is scrubbed and cleaned and placed in an oven to prevent moisture. After drying, we cut the middle part of the $5 \times 10$ cm cylindrical test body to a thickness of about 1 cm, and ground all the broken pieces to pass through a No. 100 sieve. Therefore, the ground powder in this test contains natural river sand, gravel, and cement hydration. After the completion of the composition, XRD diffraction analysis was carried out.

Figure 11 shows the XRD pattern results of the samples cured in water or $CO_2$ for 90 days. Comparing the energy peaks of the four elements, it can be found that the control group contains calcium hydroxide, calcite, and quartz, while aragonite and vaterite did not appear; the six specimens after carbon curing in this experiment were L-50-1-4, L-100-6-4, L-100-6-2, L-50-6-4, L-50-6-2, and L-50-6-8. The most relevant peaks in this study were related to calcite and calcium hydroxide, and there were some large peaks related to quartz that were attributed to sand particles within the powder. The samples have calcium hydroxide peaks at 18°, 34°, and 47° which are hydration products, however, in the carbon cured sample the pattern does not show less calcium hydroxide and is detected at 29° and 39° stronger calcite peak. The chromatograms of these six specimens are very similar, and there is no obvious difference due to the different combinations of carbon curing environments, and even the chromatograms are highly similar to those of the control group. They still contain calcium hydroxide, calcite, and quartz, but there is no signal of aragonite and vaterite. It is reasonable to explain that the XRD chromatograms of the above seven samples are very similar, and there is no obvious difference due to their low neutralization degree.

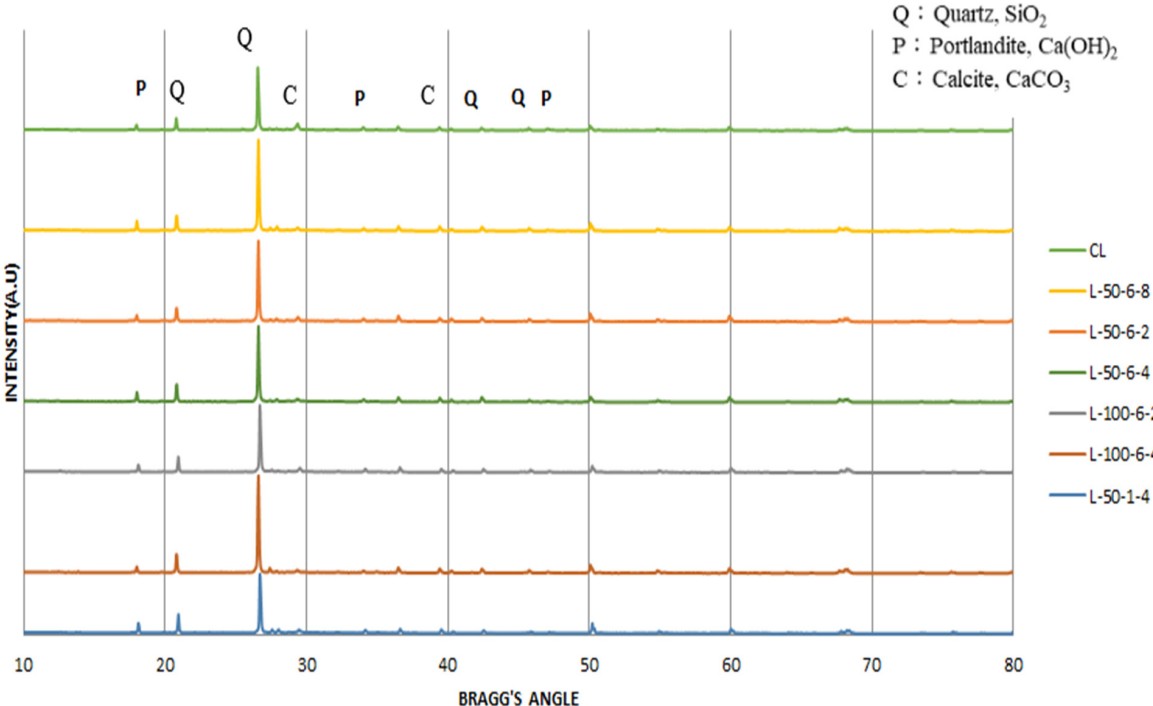

**Figure 11.** XRD profiles for the specimens at 90-day water or $CO_2$ curing (Q: Quartz, $SiO_2$; P: Portlandite, $Ca(OH)_2$; C: Calcite, $CaCO_3$).

### 3.6. Carbonation Depth

In this experiment, the $10 \times 20$ cm cylindrical specimen was cured for 90 days and placed in an oven to remove moisture. After cutting the section, a spray phenolphthalein indicator was used to observe whether the periphery was neutralized, as shown in Figure 12a,b, $CO_2$-curing, and water-curing specimens, respectively. The neutralization depth was observed with the Dino-Lite handheld digital microscope, and the neutralization depth was calculated and analyzed with the microscope software. Table 9 shows the top five neutralization depths of normal concrete $CO_2$-cured specimens, but the highest neutralization depth only penetrates 1.71 mm, and the neutralization degree is only 6.68%. This also shows that the chromatograms of the above seven specimens are very similar, and there is no obvious difference due to the low neutralization degree.

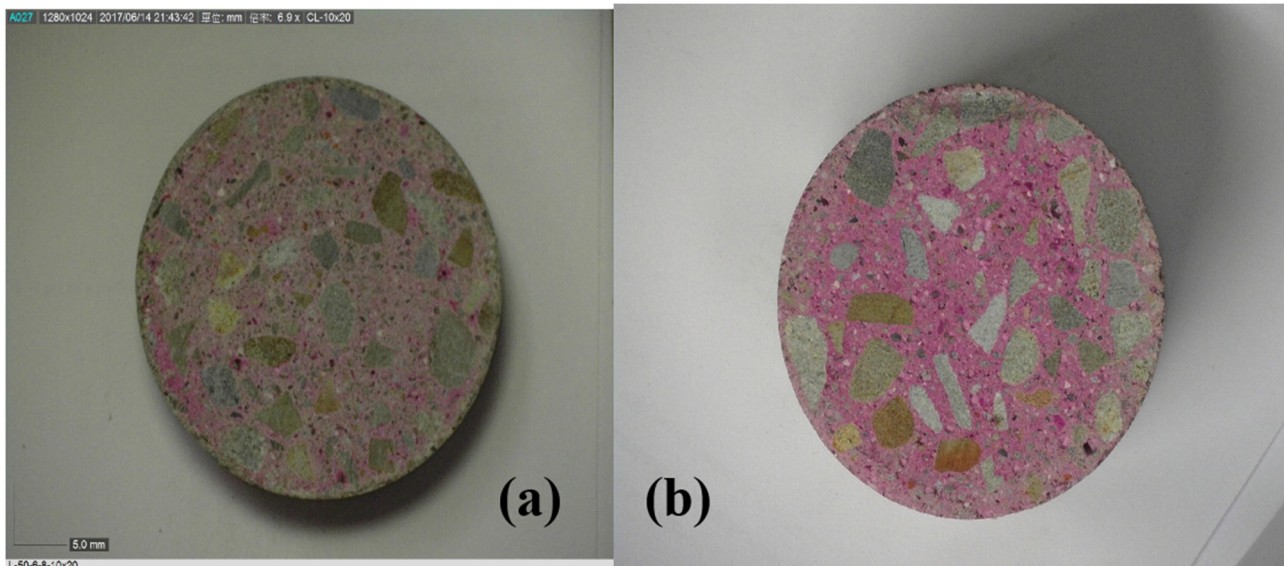

**Figure 12.** Neutralized degree observation of normal concrete (**a**) $CO_2$-cured specimen and (**b**) water-cured specimen at 90-days.

**Table 9.** Neutralization results of $10 \times 20$ cm cylindrical specimen.

| Specimen ID | Carbonation Depth (mm) | Neutralization Degree (%) |
|---|---|---|
| L-50-6-8 | 1.71 | 6.68 |
| L-75-3-4 | 1.36 | 5.38 |
| L-100-3-4 | 1.35 | 5.33 |
| L-75-3-2 | 1.29 | 5.03 |
| L-75-6-4 | 1.22 | 4.67 |
| CL (water curing) | 0.00 | 0.00 |

## 4. Summary

The main findings of this study regarding $CO_2$ curing on the mechanical properties of Portland cement concrete are summarized as follows:

(1) The early 3-day compressive strength of $CO_2$-cured concrete is higher than that of conventional water-cured concrete. When the age reaches 28 days and 90 days, the compressive strength of $CO_2$-cured concrete is close to that of conventional water-cured concrete. The strength of normal and high-strength concretes increases with concrete age, regardless of $CO_2$ concentration, duration, and pressure. The curing combination of 50-3-2 and 50-1-2 both won 3 first places in compressive strength, indicating that the $CO_2$ pressurized 0.1 to 0.3 MPa concentration of 50%, and two hours of $CO_2$-curing could get better strength.

(2)    A 5 × 10 cm cylinder of normal-strength or high-strength concrete is significantly stronger at each age than a 10 × 20 cm or 15 × 30 cm cylinder. Furthermore, the difference in strength between the 5 × 10 cm and 15 × 30 cm cylinders is large, with ks values as high as 1.18. However, the results showed no difference between 10 × 20 cm and 15 × 30 cm cylinders in normal strength concrete.

(3)    Compared with conventional water-cured concrete, the elastic modulus of carbon dioxide-cured concrete also increases continuously in proportion to the square root of the compressive strength. Choosing the most appropriate type of $CO_2$ curing for fresh concrete will have a significant impact on the modulus of elasticity.

(4)    There is only a minor difference in four $Ec$ empirical equations obtained from 5 × 10 cm or 10 × 20 cm cylinders by $CO_2$ or water curing, whereas the 10 × 20 cm cylinders obtained larger elastic modulus coefficients. The average COVs of the elastic modulus $Ec$ from 5 × 10 cm and 10 × 20 cm cylinders are 10.1% and 9.2%, respectively.

(5)    The results of neutralization depth showed that the highest neutralization depth of carbon curing specimen only penetrated 1.71 mm, and the degree of neutralization was 6.68%. This also shows that the chromatograms of the seven $CO_2$-cured samples are very similar, and there is no obvious difference due to the low neutralization degree.

**Author Contributions:** Investigation, Y.-C.W. and S.-H.K.; Writing—original draft, M.-G.L. and Y.-C.K.; Analysis, or interpretation of data—M.-G.L., W.-C.W. and H.-W.C.; Writing—review & editing, W.-C.W. and H.-W.C. All authors have read and agreed to the published version of the manuscript.

**Funding:** This research was funded by [Ministry of Science and Technology of Taiwan] grant number [107-2221-E-324-010-MY2 and 110-2625-M-008-015].

**Institutional Review Board Statement:** Not applicable.

**Informed Consent Statement:** Not applicable.

**Data Availability Statement:** Not applicable.

**Conflicts of Interest:** The authors declare no conflict of interest.

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
