# Peer review of "CO2 Curing on the Mechanical Properties of Portland Cement Concrete"

_buildings, doi:10.3390/buildings12060817_

Round 1
Reviewer 1 Report
The paper, without any doubt, is of interest from a scientific and practical point of view. However, its preparation is not properly made and requires to be improved. After some, below listed, corrections are made, the paper can be recommended for publication.
Some drawbacks to be improved:
- Page 1. The evidence of carbonation is confirmed not by C3S and C2S, as stated by the authors, but by their hydration products.
- Page 2. The following abbreviations, these are: C3S and C2S, CxSHn and Cx-1SHn, in the equations 1, 2, 3, 4 should be replaced by corresponding chemical formulas.
- Page 4. The lines in Table 3 should be named similar to those of Table 2, namely: “Normal concrete“ and “High-strength concrete“.
- Page 6. To remove "Cured with 100% CO2” from the captions to Figures 2 and 3 . The СО2 -concentration follows from designation of the specimens.
- Page 7. The expression “Normal Concrete” in Table 4 should be replaced by “Age of hardening”.
- Page 9. In Table 5, “Concrete strength/ Duration” should be replaced by “Type of concrete/age”, “Normal strength” by “Normal concrete”.
- Page 10. In Figures 7 and 8, the following values: “Ec“ and “√f'c” are absent. Moreover, designations of the specimens should correspond to those of the examples in Fig. 4 and Fig. 5.
- Page 13. In Table 7, Column “Specimen ID”, designations of the specimens should correspond to those of the examples in Fig. 4 and Fig. 5.
- Page 14. Figure 10 and Table 8 do not specify the specimens from which concrete were tested.
- Page 15. Figure 11 should be removed, because it does contain any information.
- Page 16. Table 9 does not specify a concrete type – “Normal concrete” or “High-strength concrete”. Figure 12 should be removed, because it does contain any information.
- Page 17. Conclusions should be amended by the recommendations on a choice of optimal values and parameters of the СО2-сuring: concentration, pressure, duration.
Reviewer 2 Report
Dear authors,
This is very popular topic at the moment and it has a high interest in reading. There is some good content here, however it still needs improvement.
Abstract:
L 2-3, and L 15-16: fix the font
Intro:
L 5 - This results- these results
L 3-6 change font
Section 2.1
Last line, what does it mean with estimated additional 15%? What is it referred to?
Section 3.1: Fix the font please throughout the entire section.
Improve figures 2 and 3, it is not clear what is what, maybe make it colorful.
Table 4: What does CL mean in the table?
I do not think TGA method is useful in CO2-cured concrete, it is difficult to differentiate carbonates.
Also, XRD method in this case is only qualitative, there is not much conclusion from it.
Explanation of methods and presentation of results needs some improvement.
Language needs improvement.
Reviewer 3 Report
The paper is an inquiry into the topic of CO2 curing effect on properties of concrete, in context of different sample sizes and different cruing conditions. The design of the experiment is done well, and the tests conducted are interesting. However there are a few issues I would like Authors to take into consideration:
1. Introduction does not convey well the scope of the research, its aim, and novelty of the research. Effects of carbonation on the mechanical properties are well known, just as the fact that sample size affects the test results. If Authors could add some clarifications and information about scope and novelty, it would be better for the understanding of the paper. To clarify - the scope is described in the description of concrete samples, however this is very non-intuitive approach for the reader.
2. Please add description of the coarse aggregate, as it is only described as 'crushed stone'
3. It might be beneficial to describe the curing process in more detail, as in when the sampels were inserted into the chamber, what were the water curing conditions, etc. Currently it is not clear what was the timeline of curing conditions.
4. In relation to data shown in fig 7 i 8: there is no description of what is the origin of the reference line. Moreover, I would say that it is questionable if the data obtained supports the concluision that the data follows reference line, especially in fig. 8. for CL 10x20. Could Authors provide additional information, for example trend lines with the R-squared provided to better support their description?
5. While it is good that XRD tests were carried out, why the sampling method was chosen, which is considered unfit by the Authors?
6. The paper while describes the obtained relationships, does not delve into the descriptions of why this effects happen, nor do they connect with each other or similar research, especially in case of the compressive strength.
7. Strength charts are hard to read, please consider changing the formatting.
Round 2
Reviewer 3 Report
The paper and answers provided by the Authors were satisfactory, and the paper is improved.
Author Response
Ok and thanks.